# Robot-Assisted Rehabilitation Architecture Supported by a Distributed Data Acquisition System

**DOI:** 10.3390/s22239532

**Published:** 2022-12-06

**Authors:** Arezki Abderrahim Chellal, José Lima, José Gonçalves, Florbela P. Fernandes, Fátima Pacheco, Fernando Monteiro, Thadeu Brito, Salviano Soares

**Affiliations:** 1Research Centre in Digitalization and Intelligent Robotics CeDRI, Instituto Politécnico de Bragança, 5300-252 Bragança, Portugal; 2Laboratório para a Sustentabilidade e Tecnologia em Regiões de Montanha (SusTEC), Instituto Politécnico de Bragança, 5300-252 Bragança, Portugal; 3Engineering Department, School of Sciences and Technology, UTAD, 5000-801 Vila Real, Portugal; 4INESC TEC—INESC Technology and Science, 4200-465 Porto, Portugal; 5Faculty of Engineering, University of Porto, 4200-465 Porto, Portugal; 6IEETA—Institute of Electronics and Informatics Engineering of Aveiro, 3810-193 Aveiro, Portugal

**Keywords:** rehabilitation robotics, upper limb, electromyography sensor, UR3, data acquisition, graphical user interface

## Abstract

Rehabilitation robotics aims to facilitate the rehabilitation procedure for patients and physical therapists. This field has a relatively long history dating back to the 1990s; however, their implementation and the standardisation of their application in the medical field does not follow the same pace, mainly due to their complexity of reproduction and the need for their approval by the authorities. This paper aims to describe architecture that can be applied to industrial robots and promote their application in healthcare ecosystems. The control of the robotic arm is performed using the software called SmartHealth, offering a 2 Degree of Autonomy (DOA). Data are gathered through electromyography (EMG) and force sensors at a frequency of 45 Hz. It also proves the capabilities of such small robots in performing such medical procedures. Four exercises focused on shoulder rehabilitation (passive, restricted active-assisted, free active-assisted and Activities of Daily Living (ADL)) were carried out and confirmed the viability of the proposed architecture and the potential of small robots (i.e., the UR3) in rehabilitation procedure accomplishment. This robot can perform the majority of the default exercises in addition to ADLs but, nevertheless, their limits were also uncovered, mainly due to their limited Range of Motion (ROM) and cost.

## 1. Introduction

Rehabilitation appears to be one of the most appealing and effective sub-fields of healthcare for the robot’s application. A typical rehabilitation session includes four steps: identifying the patient’s condition, defining the rehabilitation goals, planning and implementing the treatments and, lastly, evaluating the outcomes [1]. There are two types of rehabilitation: passive rehabilitation and active rehabilitation. Passive rehabilitation is employed in the early stages of rehabilitation when the patient cannot provide the necessary force to move his arm. Thus, the physiotherapist performs the movement and provides the required force to complete the exercise. Passive therapy offers no meaningful improvement in later phases [2] and is hence pointless. The physiotherapist moves to active rehabilitation, where the patient must perform most of the exercises independently.

Robots have the ability to transform and improve this sector by lowering the physical therapist’s burden, allowing less supervision by therapists, and offering more precise tracking of patient development throughout the sessions. Application with robotics in the rehabilitation process is a challenging task, as a consequence, the development of robotic technology designed to help patients perform rehabilitation is one of the most lagging sub-fields in medical robotics research and experiences many technical hurdles. The main challenges encountered include: cost, safety, efficiency, and ease of use for both the user and therapist [3].

Nevertheless, Human–Robot Collaboration (HRC) is seen as the solution in a variety of areas, including the medical field. In these fields, robots are seen as the answer to achieving high efficiency and excellent results in several domains. Robots in healthcare have emerged as a result of the various issues that these robots may overcome, which can be divided into two categories:**Limiting viral transmission:** Since the SARS-CoV-2 outbreak, the use of robots in hospitals has grown significantly, mostly to limit the spread of viruses [4]. These autonomous systems have the benefit of having intrinsic immunity to viruses, with minimal risk of disease transmission via human–robot–human contact. This ability is extremely valuable for pandemic control, as the robot may be used for cleaning, transportation, and telemedicine [5].**Patient monitoring and pressure relief:** Using onboard and/or external sensors, highlighting improvement or deterioration in the patient’s health, and can even perform some diagnostics for the most advanced robots. In addition, the robotic alternative can allow clinicians to be freed from laborious and repetitive tasks.

The rehabilitation robotics era started in the 1990s [6,7,8]; since then, a multitude of research teams have designed robots specifically for rehabilitation. Some robots are dedicated to upper limb rehabilitation [9], others are designed for lower limb rehabilitation [10], and a few devices can carry the rehabilitation for both of them [11,12]. Some of these reported robots operate in a two-dimensional (2D) environment [13]. In contrast, others operate in a three-dimensional (3D) environment with 3 Degrees of Freedom (DOF) [14], 4 DOF [15], 5 DOF [9,16], 6 DOF [12], and a few even reached 7 DOF or more [17]. Some research teams have focused on developing algorithms for using dedicated commercial robots in the industrial domain [18,19]. As an example of a developed rehabilitation robotic, ANYexo [20,21] is an upper limb rehabilitation exoskeleton robot designed with 6 DOF and a six-series elastic actuator; it has an improved design that allows for a wide range of motion.

The completion of rehabilitation sessions directly after an incident are very beneficial and can be paramount to a patient’s chances of recovery—the longer a patient delays in rehabilitation sessions, the more their chances of recovery decrease [22]. Unfortunately, the field of rehabilitation is becoming increasingly overburdened [23], and this ever-increasing number further increases the shortage of this service in some regions, particularly where a shortage of physiotherapists is observed and delays access to this service. This makes it increasingly urgent to standardise rehabilitation robots as a reliable tool to be used by physiotherapists. Nevertheless, most of the reported robotic rehabilitation solutions are still in the testing stage and their biggest challenge is to enable them to reach mass production and mass customisation capabilities [24], in order to make them accessible to a wider audience and reach as many people as possible.

Cobots, such as the UR3 and LBR iiwa from Universal Robots and KUKA, respectively, might represent a solution for a fast implementation of these technologies in the rehabilitation sector. These robots are renowned for being fast, easy to program, flexible, and safe [25], as well as robust and accurate thanks to the various types of sensors (torque, force, and current) [26] included in each of their six joints. One of their most intriguing aspects is their HRC, which is based on the ISO/TS 15066:2016 technical specification for safety criteria [27], allowing the robot to work in a stochastic and unpredictable environment. Because of these characteristics, they have received much attention from the industry and the research community. Nevertheless, one of these robots can have a potential cost of more than EUR 250,000 [28].

Therefore, the objective of this work is to promote the application and implementation of industrial robots in functional rehabilitation by employing an association of different wireless sensors connected to a single software that is able to offer, through a graphical interface, the possibility for the physiotherapist to select or easily create a rehabilitation strategy according to the needs of the patient, to have the possibility to see the effects of the rehabilitation in real time through the sensors used, and to follow the patient’s progress throughout the sessions thanks to the data saving capabilities. According to the IEC/TR 60601-4-1, defining the DOA of medical electrical equipment and described in [29], the proposed architecture has a DOA of 2.

In addition to this introduction, this paper is organized as follows: Section 2 presents the related works in the literature. Section 3 outlines the system’s overall architecture, describing the SmartHealth software, safety, and control strategies. Section 4 highlights the findings of four rehabilitation tests in graphical form, with varying constraints and settings. Finally, Section 5 concludes with a general conclusion about using UR3 for rehabilitation and in future studies.

## 2. Related Works

In this section, a closer look will be taken at some of the recently published works, starting with [30]. The design of a robotic structure for the rehabilitation of the lower limbs is presented, along with a description of the architecture allowing telerehabilitation. The proposed device focuses on people and tries to make them comfortable during its usage. It also seeks to minimize the workload of the doctors performing the rehabilitation. The work proposes a design to be easy to use, thus allowing anyone to help the person while conducting the sessions, even relatives at home. The device is provided with a remote control system with a graphical interface that helps the doctor understand the patient’s rehabilitation training status.

Further, [31] disseminates the results of a 3D video game-based rehabilitation combined with a 4 DOF Barrett WAM Robotic Arm. This system is provided with a Graphical User Interface (GUI) that permits interaction between doctor, robot, and patient. The work in [32] describes a robotic exoskeleton focused on elbow rehabilitation. This proposed solution offers remote control and sensing via the internet through the application of a Message Queue Telemetry Transfer (MQTT) protocol, as well as graphical feedback of the patient EMG. The solution implements a GUI developed through LabVIEW. The authors further propose integrating a pain estimation technique based on a cascading fuzzy decision system that enhances the security aspect of such a robotic device and promotes a safe HRC.

The same research team, led by Bouteraa, has presented in [33] a wrist rehabilitation device with a similar concept to the one given before, in addition to implementing a fuzzy classifier to estimate the patient’s muscle fatigue and wrist health status in three steps. Ref. [34] briefly presents an Internet of Things (IoT) approach architecture to connect patients, doctors, therapists, and robot administrators. The work offers several features, mainly training data, evaluation reports, managing patient information, viewing rehabilitation prescriptions, and offers real-time communication.

In [35], a design of an upper limb rehabilitation robot is presented, with a specific purpose to make the interaction between the patient, robot, and doctors closer through the development of an external sensor system that can perceive the rehabilitation environment, extract relevant information, and ensure the patient’s safety. This system comprises a force sensor, vision sensor, auditory sensor, proximity sensor, and an EMG sensor that sends the data to a so-called “sensor information coordination manager”. The proposed architecture also seeks to improve the patients’ interest and, consequently, the rehabilitation effects.

The work in [36] further proposes the dissemination of rehabilitation technologies to the general public by developing a low-cost hand rehabilitation device based on Arduino. In this way, the design of a unified platform with a relatively simple architecture allows for the connection of different devices. The currently published proof-of-concept is capable of supporting three other essential rehabilitation devices and relies on the community to extend this idea. The exercise is created by the medical staff in the form of a simple program, with a guarantee offered by the developers on the constant continuity of the programming language over the versions.

According to an analysis of the previously mentioned articles, the GUI is implemented in all of the proposed architecture except for [18,19,35]; the EMG sensor is not implemented in the majority of the discussed papers [18,19,30,31,34,36,37]. Furthermore, there is a discrepancy between the rehabilitation strategies, none of the systems presented are able to offer both the possibility to choose between default exercises and the possibility to create exercises and adapt, i.e., either to follow a pre-defined rehabilitation pathway or to create an exercise that the robot will reproduce. In addition, it is common to find in the literature studies describing the use of larger versions, namely UR5 and UR10, as rehabilitation robots. Nevertheless, some works use UR3 as a development platform. These works are mainly those of Fernandes et al. [18,19], where a simulation for rehabilitation using UR3 was proposed, and a Reinforcement Learning algorithm was implemented to provide resistance force based on the patient’s needs. In [37], a study of rehabilitation using virtual reality was performed using the UR3 to provide physical support and haptic feedback.

## 3. Rehabilitation System Architecture

The proposed system is composed of several components that communicate with one another and allows the rehabilitation treatment to be completed. Figure 1 depicts the system’s implementation. The following elements list the point indicated in Figure 1; the points listed from ***a*** to ***e*** refer to the elements applied, while the points numbered from ***1*** to ***9*** correspond to the data transmitted. The different sensors are implemented with a “plug and play” strategy.

**(a) SmartHealth software:** The software can be installed on any computer using the windows operating system. Several real-time loops are responsible for data reading, displaying, and taking action inside the software;**(b) Router:** In the case of wireless communication, a router linked via Ethernet cable to the UR3 robotic arm is necessary. The physiotherapist needs only to connect to the computer with the router network. This router act as a bridge to transmit the information between the robot and the computer. Nevertheless, a direct Ethernet connection can also be used, allowing one to ride off the router;**(c) Control box and Teach pendant:** The teach pendant is generally applied to communicate easily with the robot using the UR programming interface. Through this architecture, the teach pendant is useless. Nevertheless, the control box is still required;**(d) UR3:** Universal Robots is the robotic arm responsible for performing upper limb rehabilitation and driving the patient’s hand;**(e) Robotiq FT300:** It is an external 6 DOF force and torque sensor that allows monitoring of the patient’s force during the rehabilitation sessions;**(f) EMG sensor:** The EMG sensor allows real-time monitoring and transfer of the patient’s muscular activity via Bluetooth.

Regarding the description of the communication data:

(**1**): The physiotherapist assigns the rehabilitation method to be performed and introduces the patient’s characteristics and specificities;(**2**): Patient and exercise monitoring via the graphical software interface;(**3**): Robot command. The command sent can be a moving command, composed of a specific position/speed in $R^{6}$, followed by other parameters such as arm acceleration, arm speed, and blending with the previous set-point. It can be a data request, for example, the actual robot position, force, torque, etc.;(**4**): The data received following the requested command in (**3**). It can be an acknowledgment of the requested command, force, torque, position, or joint angle data;(**5**): Bi-directional data transmission between the router and robotic arm control box;(**6**): The continuation of the data sequence transmitted in (**3**);(**7**) and (**8**): The origin of the data transmitted to (**4**);(**9**): A bi-directional data transmission, consisting of Bytes sent to/received from the EMG Shimmer sensor.

### 3.1. SmartHealth Software

The SmartHealth software is applied to control the robotic arm. The SmartHealth program is a non-commercial, still-in-development software that provides a graphical user interface for physical therapists and enables easy robotic arm operation. This software can be used without any specific training. The current beta version of the program has a plethora of capabilities that will allow passive and active rehabilitation [38], as well as tracking throughout various sessions for a given patient. However, it does not provide a conclusion on patient improvements. The program, developed in Python, can handle the repetitive activities that serve as the foundation for standard rehabilitation sessions. Indeed, it has been demonstrated that a more significant number of repetitions results in a faster recovery [39]. The SmartHealth GUI is presented in Figure 2.

### 3.2. Control Flowchart

The complexity of the designed robots reported in the literature usually required a trained and qualified physician to perform rehabilitation using these devices; in some cases, it is even referred to the need for the patient to undergo specific training to be able to follow the rehabilitation [32]. The designed architecture allows easy control of the robot with only a few clicks and high-precision data collection thanks to the interconnection of several devices. This approach, based on commercialized devices, will promote the diffusion of these technologies on the one hand and decrease the workload of health professionals. Figure 3 represents the first part of the remote system operation.

By turning on the robot and the EMG device and getting all of them interconnected via the same network and Bluetooth to the software, the physiotherapist can first access the software database regrouping all the patients that have used the robot rehabilitation previously. In case a new patient intends to perform the rehabilitation, the healthcare servant has the possibility to introduce the patient into the database by defining the name, middle name, surname, date of birth, patient number, and the rehabilitation objectives as shown in the previous Figure 2a.

Doctors can define the treatment plan for a specific patient, which essentially means selecting an already defined exercise or creating a new custom exercise. They can choose the type of rehabilitation (passive or active) and determine the restrictions to be applied to the robot. Once everything has been defined, all devices remain in standby mode until the instructions to start the exercise are initiated. In contrast to some of the solutions proposed to record a personalized exercise (such as using cameras and the application of recognition algorithms), the physiotherapist can directly move the robotic arm without significant effort. During this movement, the software records the path that the robot passes and ensures a high fidelity of path duplication. Figure 4 illustrates the second part of the flowchart. Moreover, the following figure (Figure 5) demonstrates how the physiotherapist can carry out the custom exercise.

Depending on the characterization made earlier, the program will launch one of the modules depicted in the above figure. The physiotherapist can view the patient’s data in real-time, including force, robot position regarding the reference, and muscular activity. Thus, it allows the medical professional to assess the patient’s condition in real-time and reassess a strategy for the rehabilitation sessions. The reader should be aware that, due to the significance of the data reception frequency, the control part, the data reception and saving part, and the data display part are all carried out in parallel. In the current version, the data are gathered with an average frequency of 45 Hz.

### 3.3. Safety Strategy

Safety is the most crucial consideration during rehabilitation sessions; as robots interact so closely with individuals, they might result in harming the users. In this regard, the robot must not exceed the patient’s ROMs to prevent injuries. The ROMs can be pre-programmed in advance by restricting the robotic arm’s movements. Furthermore, as for the UR5, the UR3 is equipped with an internal safety system that terminates all robot movements if the force surpasses 250 N within 500 ms, which is ideal for HRC. The ROBOTIQ FT300 force/torque sensor, described below, can also be employed to measure the force released by the patient in real-time; this value can be configured to 50 N for force and 8 Nm for torque according to [40]. Actually, the safety measures described above are sufficient for a patient who has recovered most of his muscle strength and is in the last phase of his rehabilitation.

### 3.4. Control Strategy

The robot control varies according to the rehabilitation type selected, with the software allowing the user to choose between passive and active-assisted rehabilitation. The UR3 internal control algorithm is also employed to enable the Tool Center Point (TCP) to reach the exact position requested based on a reference sent and completes the motion control. However, this reference varies based on the type of rehabilitation. The UR3 is equipped with a force–torque sensor, as illustrated in Figure 1. It is a six axes sensor with specifications presented in Table 1. The sensor is fitted between the robot and a 3D-printed custom arm handle (see GitHub repository https://github.com/RahimCHELLAL/SmartHealth). The data $F_{px},F_{py}$ and $F_{pz}$ represent the measure of force in different axes. The variables $M_{x},M_{y}$ and $M_{z}$ are the moments that can be measured.

For passive rehabilitation, the software sends the reference position (x,y,z) in the robot’s Cartesian frame and the TCP orientation angle $\left(\theta_{x},\;\theta_{y},\;\theta_{z}\right)$ to the robot over a Transmission Control Protocol/Internet Protocol (TCP/IP), followed by the desired acceleration, velocity, and blending with the previous point for a smoother transition.

For assisted active rehabilitation, the reference is the robot arm speed. The $\dot{q}$ is the reference speed in $R^{6}$, represented by the $\left(\dot{x},\;\dot{y},\;\dot{z}\right)$ linear velocities and $\left(\dot{\theta_{x}},\;\dot{\theta_{y}},\;\dot{\theta_{z}}\right)$ angular velocities. The actual speed is extracted using the sensor force data $F_{p,i}=(F_{px},\;F_{py},\;$$F_{pz},\;M_{x},\;M_{y},\;M_{z})$ from the patient’s hand. Figure 6 shows the control block diagram for assisted active rehabilitation in a general sense.

It should be noted that, for the control in *y* and *z* axis for the 2nd and 3rd exercise described in the Results and Discussion section, the error between the actual position (*q*) and reference trajectory is also added to Figure 6; this error ($\varepsilon_{q}$), multiplied with a specific gain $\left(P^{\prime }\right)$, is then subtracted to the patient’s hand speed ($\varepsilon_{\dot{q}}\cdot P$). When the position error is too high, the robot will try to decrease this error and does not take into consideration the patient’s movement. Figure 7 represents the block diagram in *y* and *z* axis, used for the 2nd and 3rd exercise.

$F_{i}=\left(F_{ix},\;F_{iy},\;F_{iz},\;M_{ix},\;M_{iy},\;M_{iz}\right)$, represent the proper force applied by the robot to move the patient’s arm. $q_{i}=\left(x,\;y,\;z,\;\theta_{x},\;\theta_{y},\;\theta_{z}\right)$, represent the TCP position. The reference position f(x,y,z) is the mathematical equation describing the reference trajectory.

### 3.5. Shimmer EMG Device

EMG sensors can monitor the electrical activity of a muscle for a given moment. This information can be essential during rehabilitation, as it can provide physiotherapists with vital clues and the ability to interpret the response, positive or not, of the muscle to the chosen treatment [42]. SHIMMER is an exceptionally extensible wireless sensor platform that can be used for biomedical research applications. The sensor’s small size and light weight (20 g) make it ideal for body-worn kinematic and physiological sensing applications [43] and can reach a sampling frequency of 250 Hz. Every SHIMMER’s data can be accessed on-the-fly through the usage of the Consensys Software streaming capabilities over Bluetooth. The software also permits further offline data processing. The manufacturer describes the Consensys software as an integrated solution for managing SHIMMER 3 devices over Bluetooth [44]. Unfortunately, the defined requirement of an on-the-fly acquisition of several data points does not permit the usage of external software.

## 4. Results and Discussion

Four exercises focused on shoulder rehabilitation were performed to assess the viability and performance of the proposed architecture and the UR3 application in the field of rehabilitation. The difference between the exercises is highlighted as follows:The first exercise is a *Lateral Shoulder Rotation* exercise in a passive mode, where it is intended for the patient to offer less or no force;The second exercise is a *Lateral Shoulder Rotation* exercise in a restricted active-assisted mode, where the robot movement is restricted only to a 2D planner environment (x,y) and $\theta_{z}$ ($R^{2}\times S^{1}$);The third exercise is a *Lateral Shoulder Rotation* exercise in a free active-assisted mode, where the robot’s movements are totally free in $R^{3}\times S^{1}$, with restriction applied only in $\theta_{x}$ and $\theta_{y}$;The fourth exercise is an ADL; the chosen exercise consists of mimicking the action of taking a cup, drinking from it, and putting it back. This exercise can be performed thanks to one of the functionalities offered by the software which is the “*Custom exercise*”. With this functionality, it is possible to manually create any exercise by simply moving the robotic arm; its movements are then recorded and reproduced.

The Lateral Shoulder Rotation carried out in most test exercises is performed as follows. The patient, while standing, bends the elbow at a right angle and moves the arm outward as far as their ROM allows in a horizontal plane while keeping their elbow fixed to their body [45]. Figure 8 shows the lateral shoulder rotation, with the blue line representing the robotic arm’s trajectory. The SHIMMER device will collect muscle activity from two different muscles in these exercises. The data acquired in channel 1 are for the deltoid muscle, while the data acquired in channel 2 are for the Latissimus Dorsi.

### 4.1. Experimental Settings

The UR3 robot is applied to perform the rehabilitation, and the robotic arm communication with the software is running at the maximum allowed frequency of 125 Hz. Nevertheless, the sampling frequency is evaluated at 45 Hz. The contact point between the patient hand and the robot is achieved through an arm handle that can be seen in Figure 8. The subject performing the exercise is a healthy 24 years old male. The same person performs all the tests reported in this article with his right hand.

### 4.2. Experimental Tests

In this exercise, the first test is the passive lateral rotation of the shoulder. As previously mentioned, the person exerts virtually no force to assist the movement, with the cobot following the pre-programmed trajectory. The following Figure 9 and Figure 10 present the gathered data for this exercise, and the EMG signal gathered for the same exercise, respectively, with a single repetition.

In Figure 9 can be seen that the robot follows the reference path perfectly. The fluctuation seen is of a neglectable value; the robot, in this case scenario, did not follow the force that was sensed. The subject undergoes the movement and as can be seen in the figure representing the patient’s force (Figure 9d) with a meager value (less than 5 N). However, a variation is noticed in the *y*-axis and *x*-axis in the middle of the exercise. This peak is the patient resistance when the robot reaches the ROM’s limit. It is essential to point out to readers that the force data gathered and shown above is a summation of the robot’s force $F_{i}$ and patient’s force, which is visible in the block diagram shown in Figure 6 and Figure 7.

In Figure 10, from both graphics (a) and (b), it can be seen that both of these muscles get activated while performing the movement and rest when the subject reaches the limit of its ROM. The graph demonstrated in Figure 10c is superficial but is added so that muscle activity can be related as a function of exercise progress.

The second exercise results are shown in Figure 11 and Figure 12 where the patients can move the robot to the desired direction in a restricted manner, only in $R^{2}\times S^{1}$.

From Figure 11, the hand movement in $S^{1}$ is hardly noticeable, as the $\theta_{z}$ movement is not printed in the graphics. Nevertheless, the robot’s movements in the (x,y) plane can be seen. At the same time, the *z*-axis remains constant with an inaccuracy of 0.0027 m with the reference trajectory. The vector of the patient’s force that is transmitted and taken into account by the robot can also be seen in the same figure; this vector represents the subtraction between the direction of the patient’s force and the error between the actual position and the reference trajectory; this strategy aims to minimize the error between the actual position and the reference trajectory, while also taking the direction desired by the patient into account. In contrast to the preceding picture, the patient’s forces involved in $R^{3}$ exhibit a more considerable variance for $F_{x}$ and $F_{y}$ (between 10 N and −10 N), whereas $F_{z}$ remains reasonably consistent around 15 N.

The results of the third exercise, where the patient’s hand is able to move freely in $R^{3}\times S^{1}$, are shown in Figure 13 and Figure 14.

The observations made for Figure 11 are also applicable to Figure 13, except for a few details regarding the movement and force on the *z*-axis. It can be seen that the robot takes into consideration the patient’s force on the *z*-axis, which was ignored in the second exercise, and moves accordingly in the third one. This force has reached values up to 20 N. The implemented controller helps the movement to be seamless and closer to the reference trajectory.

As for Figure 12 and Figure 14, more significant variance in the muscle data can be observed compared to Figure 10. This means that the choice to give the subject more freedom results in higher muscular activity. Thus, these data can provide the physiotherapist with an overview of the effort provided by the patient and can significantly influence the choice of his future rehabilitation strategy.

In a stroke rehabilitation program, strategies are primarily based on a task-oriented approach and aim to improve the patient’s movement control, range of motion, accuracy, and hand-eye coordination [39]. The tasks identified as fundamental for regaining independence in ADL were: eating, drinking, toileting, and using devices for communication (such as smartphones and computers) [46]. Thus, the fourth and final exercise presented in this article involves mimicking the action of grabbing a bottle of water, reaching for it to drink, and then putting it back down. Figure 15 shows the TCP trajectories, as well as the patient force involved in (x,y,z).

As shown in Figure 15, the UR3 can offer the necessary ROM to perform such an ADL exercise. The patient can freely exercise his arm in $R^{3}\times S^{3}$. No objects were grasped during the exercise. The graphics showing the gathered data can be saved and exported in a Comma Separated Value (CSV) file containing all the data of interest. These files can be downloaded automatically when the software is launched or forwarded to another computer. All the resulting data shown in this section are available for download in a GitHub repository (https://github.com/RahimCHELLAL/SmartHealth) with a link available in the “Data Availability Statement” section.

## 5. Conclusions and Future Work

Rehabilitation robotics essentially combines continuous interaction with the patient’s limb to accomplish therapy treatments, which is thought to be a suitable supplement to regular rehabilitation sessions. However, it still requires physical therapist supervision. Different designs and models have been reported, but a design consensus has yet to be reached. The designs differ from one robot to the next, depending on the robot constraints (DOF), size (ROM), and body part targeted (upper or lower limb).

The shortcomings of some of the existing upper limb rehabilitation robotic systems (lack of autonomy, manufacturing problems, and low sampling rate) are mainly due to the use of unitary data collection systems. This paper proposes the use of commercially available robotic arms with their integration into a simple architecture centralized by SmartHealth software capable of collecting different types of data in parallel and in real time. Additionally, this work offers patient profile management and exercise selection. The different data are monitored and recorded, thus offering a classification of 2 DOF. The software also allows the data to be exported, permitting to offer further processing possibilities by using external software such as MATLAB.

In this work, after completing the architecture design and carrying out their integration and interconnection, an HRC demonstration and a viability study of the application of the UR3 as a rehabilitation robot applying the same architecture was carried out through a preliminary study. The results of the experimental tests first confirm the viability of this architecture and the applicability of the UR3 in the field of rehabilitation; they have demonstrated all the capabilities of the robot associated with the SmartHealth software to perform upper limb rehabilitation. To demonstrate the results, a first video is published and is accessible at Youtube: https://youtu.be/TR-I_WQIs_w. A second video summarizing the project can be accessed through this link https://youtu.be/auFwzHZbiBQ (both links were accessed on 29 October 2022).

The UR3 is a mighty robotic arm and the most affordable variant of the universal robot. However, the cost of these robots is not negligible and cannot be covered by most medical institutions. In addition, while it is an excellent alternative to the larger model (UR5 and UR10), limitations have been noticed while attempting to execute some particular exercises, such as the shoulder abduction exercise with a straight elbow. This constraint is caused by the robot’s ROM being confined to a 1-m diameter sphere.

In terms of future work, it is considered to implement an additional tab or a complementary software that permit the processing of the data that were gathered through an intelligent algorithm to assess the patient’s performance during the sessions. In addition, clinical trials with the UR3 robotic arm will be conducted in a local hospital to compare its capabilities to other robotic rehabilitation projects documented in the literature and the traditional rehabilitation technique. The clinical studies will be carried out employing the same architecture, which has been proven to perform exercises while recording patient data and giving export capabilities with a relatively high frequency.

Another shortcoming observed in the rehabilitation robotics sector is the limitation of the developed software to a specific robot, and no real connection exists between each solution. Thus, further development of the SmartHealth software permits the possibility of controlling other types of robotic arms, such as KUKA and enables a multi-control aspect, allowing it to control several devices simultaneously.

## Figures and Tables

**Figure 1 sensors-22-09532-f001:**
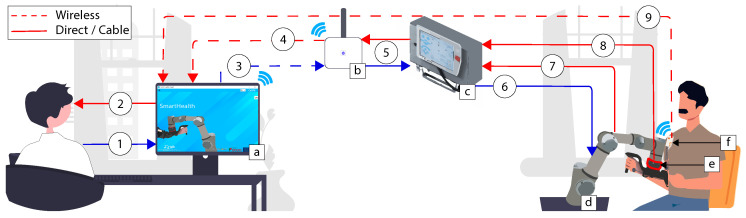
Illustration of the overall system architecture.

**Figure 2 sensors-22-09532-f002:**
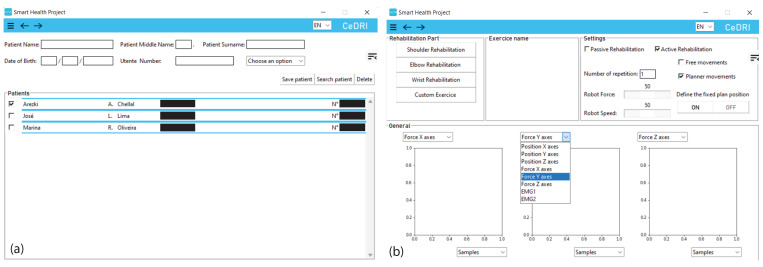
SmartHealth, control software graphical interface in an idle state. (**a**) Patient tab, with the different patient registered in the software. (**b**) Setting tab, with a slide showing the different parameters that can be plotted.

**Figure 3 sensors-22-09532-f003:**
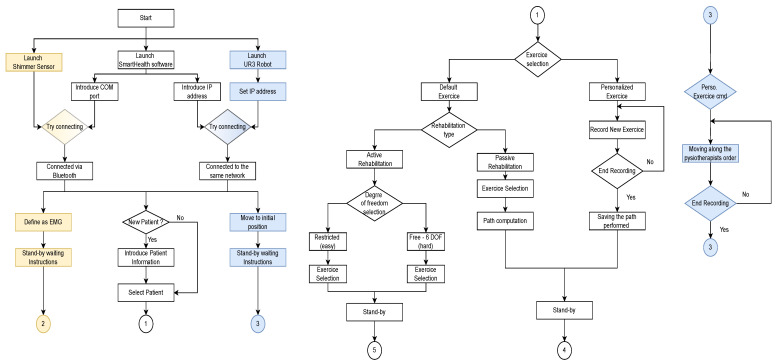
Operation flowchart for the remote control system, part 1. **White color** defines the flowchart in the software. **Blue color** defines the flowchart in the robot. **Yellow color** defines the flowchart in the shimmer sensor.

**Figure 4 sensors-22-09532-f004:**
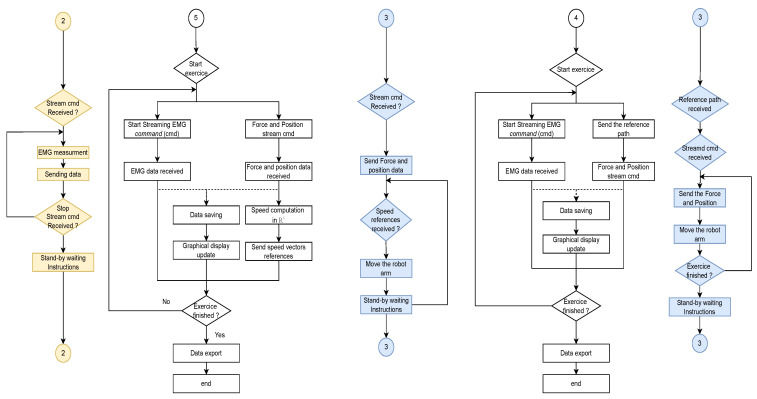
Operation flowchart for the remote control system, part 2. **White color** defines the flowchart in the software. **Blue color** defines the flowchart in the robot. **Yellow color** defines the flowchart in the shimmer sensor.

**Figure 5 sensors-22-09532-f005:**
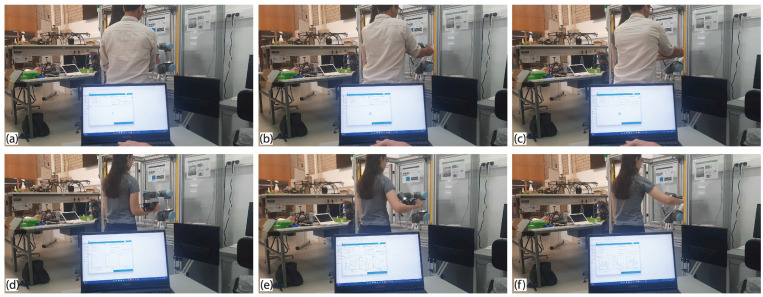
Custom exercise experimentation. (**a**–**c**) Person driving the robot and the subject’s hand, with the software recording the data. (**d**–**f**) The robot driving the patient’s hand following the exact path recorded, with the software displaying the force data in (x,y,z) on-the-fly.

**Figure 6 sensors-22-09532-f006:**
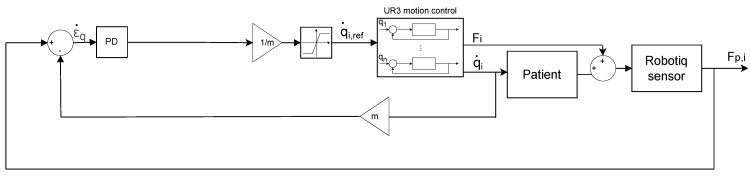
General control block diagram for active-assisted rehabilitation exercise.

**Figure 7 sensors-22-09532-f007:**
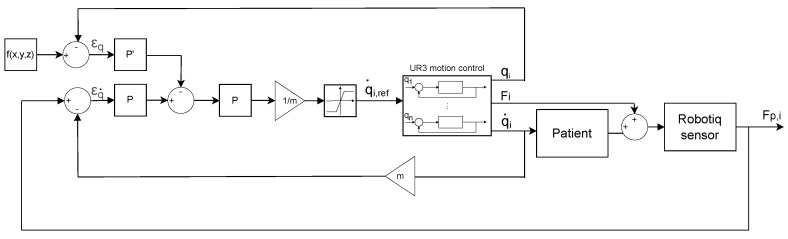
Control block diagram for active-assisted rehabilitation exercise in *y* and *z* axis for 2nd and 3rd exercise.

**Figure 8 sensors-22-09532-f008:**
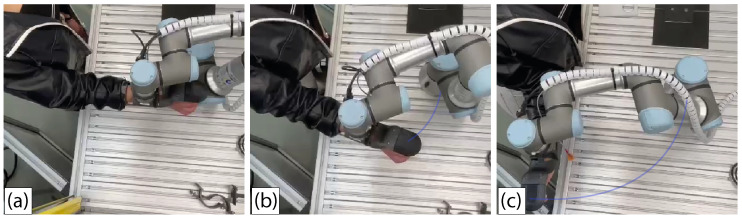
Top view of the Lateral Shoulder Rotation exercise performed by the UR3 robot. (**a**) Beginning of the repetition. (**b**) Middle of the repetition. (**c**) Reaching the limit of the patient’s admissible range of motion. **Blue line** represents the path followed by the robot [38].

**Figure 9 sensors-22-09532-f009:**
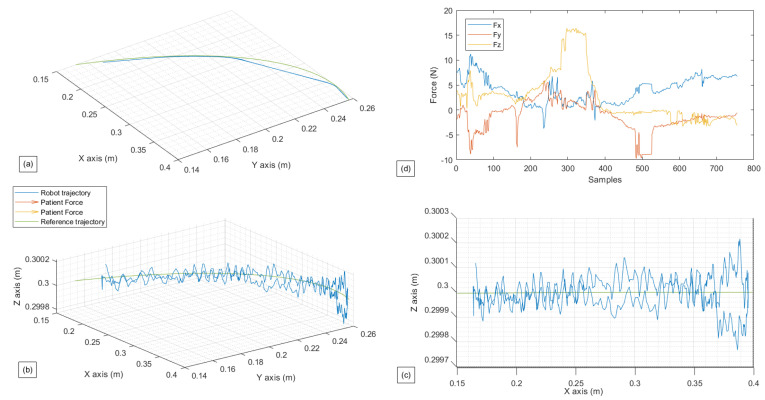
Experimental test for passive Lateral Shoulder Rotation. (**a**) Robot path performed in contrast with the reference path in (x,y). (**b**) Robot path performed in contrast with the reference path in (x,y,z). (**c**) Robot path performed in contrast with the reference path in (x,z). (**d**) Forces gathered by the Robotiq sensor in (x,y,z) in relations with the samples.

**Figure 10 sensors-22-09532-f010:**
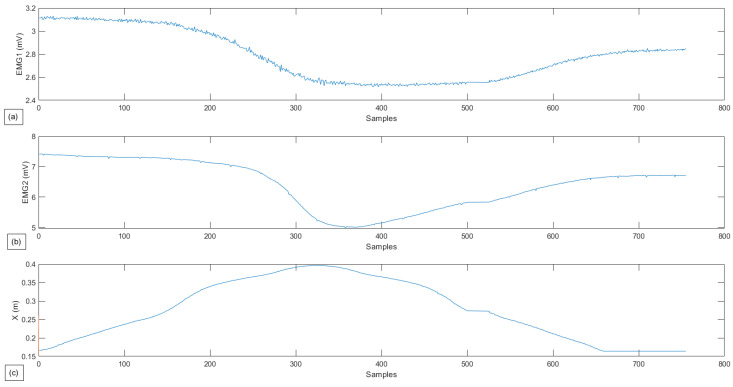
Experimental test for passive Lateral Shoulder Rotation. (**a**) Muscle activity for the Deltoid muscle. (**b**) Muscle activity for the Latissimus Dorsi. (**c**) Robot position in *x* in relation to the samples.

**Figure 11 sensors-22-09532-f011:**
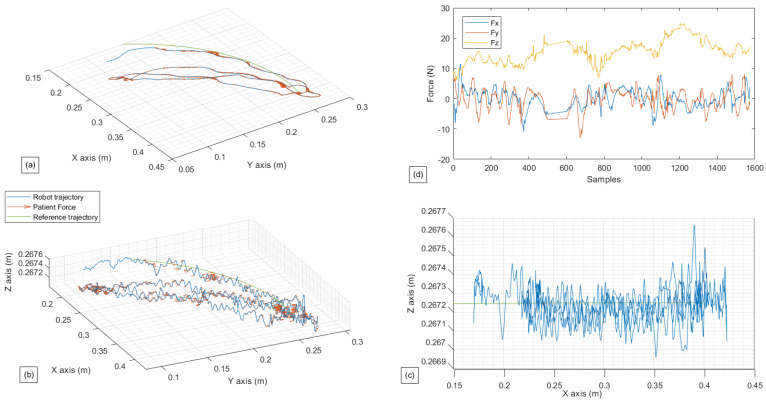
Experimental test for restricted active-assisted lateral shoulder rotation. (**a**) Robot path performed in contrast with the reference path in (x,y). (**b**) Robot path performed in contrast with the reference path in (x,y,z). (**c**) Robot path performed in contrast with the reference path in (x,z). (**d**) Forces gathered by the Robotiq sensor in (x,y,z) in relations with the samples.

**Figure 12 sensors-22-09532-f012:**
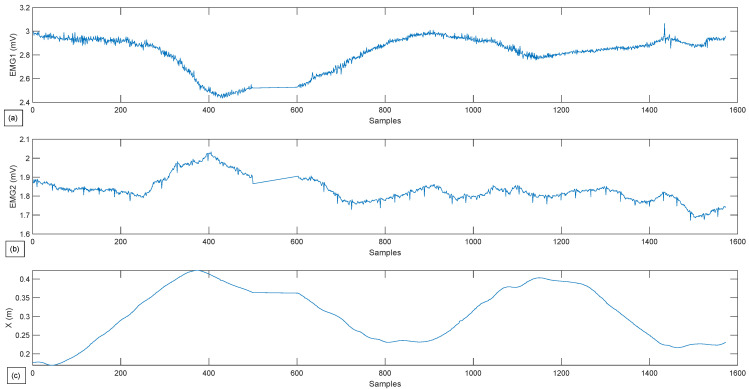
Experimental test for restricted active-assisted lateral shoulder rotation. (**a**) Muscle activity for the Deltoid muscle. (**b**) Muscle activity for the Latissimus Dorsi. (**c**) Robot position in *x* in relation to the samples.

**Figure 13 sensors-22-09532-f013:**
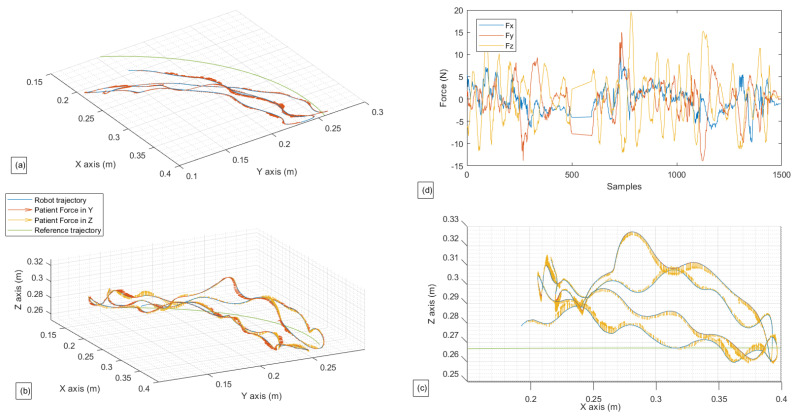
Experimental test for a free Active-Assisted lateral shoulder rotation. (**a**) Robot path performed in contrast with the reference path in (x,y). (**b**) Robot path performed in contrast with the reference path in (x,y,z). (**c**) Robot path performed in contrast with the reference path in (x,z). (**d**) Forces gathered by the Robotiq sensor in (x,y,z) in relations with the samples.

**Figure 14 sensors-22-09532-f014:**
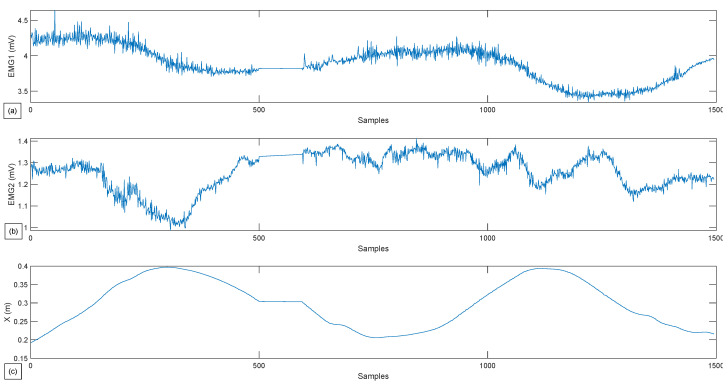
Experimental test for a free Active-Assisted lateral shoulder rotation. (**a**) Muscle activity for the Deltoid muscle. (**b**) Muscle activity for the Latissimus Dorsi. (**c**) Robot position in x in relation to the samples.

**Figure 15 sensors-22-09532-f015:**
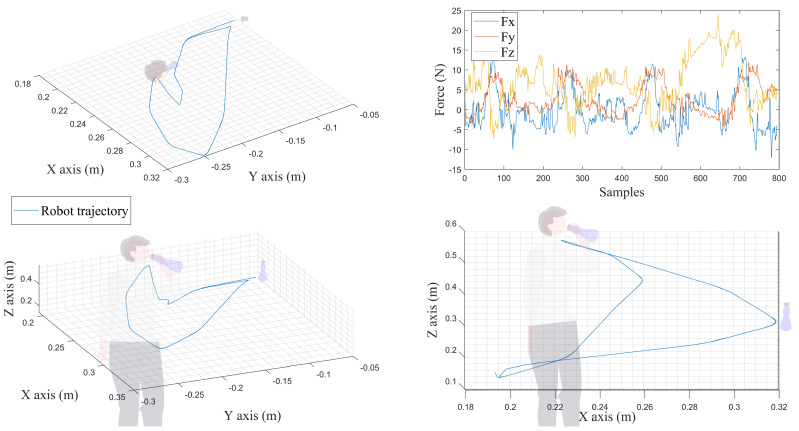
Experimental test for an ADL based rehabilitation—drinking case.

**Table 1 sensors-22-09532-t001:** Robotiq FT300 Force/Torque sensor characteristics [41].

Feature	Symbol	Value	Unit
Force Measurement Range	$F_{px},F_{py},F_{pz}$	±300	N
Moments Measurement Range	$M_{x},M_{y},M_{z}$	±30	N.m
Data Output Rate	F	100	Hz
Weight	m	0.3	kg

## Data Availability

The data presented in the Results and Discussion section is available on a GitHub repository, that can be reached via the following link: https://github.com/RahimCHELLAL/SmartHealth (accessed on 29 October 2022).

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
