# Peer review of "Robot-Assisted Rehabilitation Architecture Supported by a Distributed Data Acquisition System"

_sensors, 2022, doi:10.3390/s22239532_

Round 1
Reviewer 1 Report
In this paper, authors describe an architecture that can support upper limb rehabilitation with industrial robots, such as UR3. This study shows how small robots can perform medical applications, providing an alternative to traditional healthcare techniques. A graphical user interface is used to control the robotic arm. It was shown that the robot can perform the majority of the default exercises, as well as Activities of Daily Living (ADL), but its limitations were also revealed, mainly due to its limited Range of Motion (ROM).
The paper is well presented and deals with an attractive field. However, the reviewer would like to highlight the following points:
1. The abstract should be rewritten to focus directly on the objective of the work, the basic methodology used and the main conclusions, including the main results obtained (from the results) and the main conclusions (of Discussion).
2. Application field cannot be a key word
3. The statement of the problem should be more prominent in the first sentences of the introduction.
4. There is not enough proper substantive discussion of the problem and its significance, scope and limitations, etc.
5. Although the authors have described what has been done before by citing the relevant literature, they have not sufficiently highlighted their contribution and how their work differs from published related work.
6. SmartHealth software, it is not clear if this graphical user interface was developed in this project (the subject of the article) or if it was a previous work. Why is it called smart!!
7. This software offers patient profile management and exercise selection. However, the various data is monitored, recorded and exported, allowing it to be further processed using other software such as MATLAB. The question here is why the developed software does not have the capability to handle all the required functionality, why one needs to interfere with other software. For example, using Matlab or LabView, we can provide software that offers user interface + recording + display of curves + processing and calculation, etc.
Author Response
Dear Reviewer,
We appreciate your dedication in this review process, thank you. In the attachment are our Notes.
Best regards.

Reviewer 2 Report
Dear Authors,
We are highly appreciating for your research contribution in Robot-Assisted Rehabilitation Architecture.
1. Robot-Assisted Rehabilitation Architecture was pretty good with technical details
2. Smart Health software in line with real-time Robotic arm and Communication established with in the system was highly appreciable contribution.
3. Mechanical and sensory data will integrate in lines of Co-bot systems towards rehabilitation approaches.
Questions
1. As mentioned in line 372 &373, the proposed system having better results comparatively with other approaches in fields of
a. lack of autonomy
b. manufacturing problems
c. low sampling rate
a) Authors provide comparative analysis for above parameters
2. End user feedback is considered at real time execution.
a. What type of feedback mechanism is used in this system, kindly provide the approach.
b. In process of rehabilitation, how rigid human body shoulder conditions are resolved with proposed approach, what are them.(optional)
Author Response

(The authors gave the same response as above.)

Round 2
Reviewer 1 Report
The authors have satisfactorily addressed most of my comments and questions.